# Management of Low Anterior Resection Syndrome (LARS) Following Resection for Rectal Cancer

**DOI:** 10.3390/cancers15030778

**Published:** 2023-01-27

**Authors:** Harald Rosen, Christian G. Sebesta, Christian Sebesta

**Affiliations:** 1Department of Surgical Oncology, Sigmund Freud University, 1020 Vienna, Austria; 2Science Center Donaustadt, 1220 Vienna, Austria; 3Department of Oncology, University Clinic Donaustadt, 1220 Vienna, Austria

**Keywords:** rectal resection, low anterior resection syndrome, transanal irrigation

## Abstract

**Simple Summary:**

The present narrative review aims to give an overview about the causes and the management of low anterior resection syndrome, a common postoperative functional disorder following rectal resection.

**Abstract:**

Introduction: A total of 60–80% of patients undergoing rectal resection (mostly as a treatment for rectal cancer) suffer from a variety of partly severe functional problems despite preservation of the anal sphincter. These patients are summarized under the term low anterior resection syndrome (LARS). Preoperative radiotherapy, vascular dissection and surgical excision of the low rectum and mesorectum lead, alone or all together, to a significant impairment of colonic and (neo-) rectal motility. This results in a variety of symptoms (multiple defecation episodes, recurrent episodes of urge, clustering, incontinence, etc.) which are associated with severe impairment of quality of life (QOL). Methods: This narrative review summarizes the present state of knowledge regarding the pathophysiology of LARS as well as the evidence for the available treatment options to control the symptoms resulting from this condition. Results: A review of the literature (Medline, Pubmed) reveals a variety of treatment options available to control symptoms of LARS. Medical therapy, with or without dietary modification, shows only a modest effect. Pelvic floor rehabilitation consisting of muscle exercise techniques as well as biofeedback training has been associated with improvement in LARS scores and incontinence, albeit with limited scientific evidence. Transanal irrigation (TAI) has gained interest as a treatment modality for patients with LARS due to an increasing number of promising data from recently published studies. Despite this promising observation, open questions about still-unclear issues of TAI remain under debate. Neuromodulation has been applied in LARS only in a few studies with small numbers of patients and partly conflicting results. Conclusion: LARS is a frequent problem after sphincter-preserving rectal surgery and leads to a marked impairment of QOL. Due to the large number of patients suffering from this condition, mandatory identification, as well as treatment of affected patients, must be considered during surgical as well as oncological follow-up. The use of a standardized treatment algorithm will lead to sufficient control of symptoms and a high probability of a marked improvement in QOL.

## 1. Introduction

Following the change in surgical strategy in the treatment of rectal cancer, from abdominoperineal resection (APR) with the formation of a permanent colostomy to sphincter-preserving modalities (such as low anterior resection (LAR), ultralow rectal resection or even intersphincteric resection with coloanal anastomosis), the majority of patients eligible for radical surgery undergo a procedure leading to the preservation of the anal sphincter [1,2].

In addition to this development, multidisciplinary management introducing multimodal oncologic therapy approaches has led to a significant improvement in the prognosis for rectal cancer patients [3,4]. Due to screening programs (leading to earlier detection) as well as the above-mentioned progress in multimodal therapy, more and more patients do not only experience long-term survival, but also negative consequences associated with their individual treatment [5,6,7].

In this context, functional disorders summarized under the term “low anterior resection syndrome” (LARS) have gained increasing awareness among caretakers dealing with the treatment of rectal cancer patients in the recent past [8,9,10,11].

The following review offers an overview of the presently available evidence regarding the causes and management of LARS.

## 2. Methods

A narrative review based on literature research in Medline and Pubmed under the term “LARS” or “low anterior resection syndrome”.

### 2.1. Etiology of LARS

Pathophysiological pathways leading to the creation of LARS are not yet completely understood and remain partly unclear [12,13,14,15]. Due to the variety of symptoms associated with this condition, different etiologic factors might be responsible for its development [12,13] (Table 1).

Anal continence is an interaction of various factors including anal sphincter function, anorectal sensation, an intact reservoir function of the ampulla recti (so-called rectal compliance), stool consistency and an appropriate emptying process during defecation [14]. Due to this complex situation, every aspect of the treatment of rectal cancer can lead per se or in combination to an impairment of the physiologic function of the anorectum.

Standardized radical surgical treatment of middle and low rectal cancer using LAR with total mesorectal excision (TME) will cause a loss of the reservoir function, including a reduction in storing as well as a markedly disturbed evacuation, thus leading to a significant impairment of rectal compliance [16]. This situation is associated with an increase in false (unproductive) urge to defecate and is in strong correlation with the height of the anastomosis [17]. Karanijia and coworkers described, in a recent study performed on 232 patients, a worsened ampullary function with a decreasing distance of the anastomosis from the anal verge [18]. In accordance with their findings, a remaining rectum ≥4 cm in length was associated with significantly better functional outcomes (rectoanal inhibitory reflex, rectal capacity) compared to patients who had less than 4 cm of rectum left [19].

Since ultralow resection is sometimes unavoidable, several surgical efforts to restore reservoir function have been advocated in the past, including the formation of colonic pouches (“neoampulla”), side-to-end anastomosis or coloplasties [16]. Although published data indicate a functional (though not consistent) improvement in the first postoperative period, other publications show a loss of this benefit in the longer follow-up [16].

In addition to the loss of capacity of the rectal ampulla following rectal resection, additional pathways affecting colonic motility have been repeatedly described and seem to have an effect on the development of LARS [20,21,22,23]. Special emphasis has been placed on the impact of the so-called “rectosigmoid brake” [21,22,23]. For decades, many authors have postulated that sigmoid motility plays an essential role in delaying rectal filling, thus working as an additional “functional sphincter” to support continence function [24]. Using high-resolution colonic manometry, Dinning and others were able to show that the cyclic motor patterns (CMPs) were active in propagating into a mainly retrograde direction and increasingly postprandial [21,25]. There was scientific evidence that most of these CMPs originated in the rectosigmoid area, thus limiting rectal filling [25].

Therefore, it is more than plausible that surgical excision of the rectosigmoid colon and the effect on colonic motility could serve as an additional explanation for some of the symptoms commonly seen in LARS.

Furthermore, denervation of the parasympathetic and sympathetic nervous system has been demonstrated to lead to colonic hyperactivity and negative effects on rectal evacuation. In particular, “high ligation” at the origin of the inferior mesenteric artery seems to lead to the denervation of sympathetic nerve fibers. Koda et al. performed intraluminal pressure measurement and transit studies in a cohort of 67 patients who had undergone LAR [26,27]. When they compared a group with high ligation to a collective of patients who were treated by ligation of the superior rectal artery only, a worse functional outcome with fewer propagating contractions and increased non-propagating contractions could be observed in the group with “high ligation” [26]. However, despite these findings, there is still an ongoing debate about the real impact of the high ligation approach on functional disorders after rectal resection [28].

In addition to surgery, radiotherapy plays a major role in the multimodal management of rectal cancer. Although neoadjuvant radiotherapy has significantly improved sphincter preservation (by successful downstaging and downsizing) as well as local tumor control, the negative impact on postoperative functional results and quality of life have also been repeatedly described [29,30].

Mechanisms attributed to radiotherapy-induced damage of the rectal functions are not completely understood but changes in the colonic wall and mesentery (partly also seen in inflammatory bowel disease) have been reported and are very likely the pathogenic mechanisms [31,32].

Comparable to neuronal damage caused by surgery, neuropathy can also be a sequela of radiotherapy (mainly due to fibrosis) leading to disruption of the autonomic neural pathways [32,33]. This agrees with repeated findings of radiotherapy being a strong risk factor for LARS [29,34].

Finally, the construction of an ileostomy as protection for patients with low or ultralow rectal anastomosis is also regarded as a risk factor for functional problems after a stoma reversal [34,35,36]. Although it is undisputed that protective temporary ileostomies have a protective effect against septic complications after anastomotic leakage, the negative effect of exclusion of the colon must be taken into account. Loss of intraluminal nutrition to the colon (acetate, proprionate and butyrate) can lead to significant intestinal dysbiosis and, subsequently, mucosal atrophy [37]. Once the fecal stream starts to pass over the excluded colon, absorptive problems and inflammatory changes can occur leading to symptoms associated with LARS.

### 2.2. Symptoms of LARS

It must be accepted that LARS is associated with a large variety of symptoms, all of which, individually or in combination, can lead to a detrimental effect on patients’ quality of life (QOL). In a recent consensus paper, eight symptoms of LARS with one or more consequences were listed, as illustrated in Table 2 [34].

While many patients were mainly described as suffering from incontinence following rectal resection in the past, closer monitoring of functional outcomes following LAR has shown that incontinence is mostly a final consequence of other uncontrollable symptoms (e.g., multiple unproductive defecation episodes per day and night) [9,34,38].

Therefore, it was necessary to develop more specific scoring instruments for the identification of LARS, which do not focus on incontinence only but rather on symptoms such as urge, number of defecation episodes, etc. [39,40].

Out of them, the “LARS score”, with five simple questions and three or four answering categories, has established itself as an easy and reproducible instrument and has been translated into more than 35 languages [40]. The score has a range from 0 to 42 points, thus identifying patients with “No LARS”, “Minor LARS” and “Major LARS” (Table 3).

Although this scoring system offers a valid approach to identifying and following patients with LARS, some weaknesses must be taken into account (i.e., lack of information about consequences on QOL, impairments of other organ functions such as sexual activity, bladder emptying, etc.) [34].

Once patients suffering from LARS have been identified, further diagnostic processing will have to rule out organic lesions (anastomotic stenosis, local recurrence, side effects from radiotherapy) as well as changes in stool consistency (diarrhea) as reasons for the functional problems reported by the patient. Anorectal examination with digital inspection as well as endoscopy will be able to clarify these questions in a fast and cheap manner.

More sophisticated tests, such as anorectal physiology evaluation (anal manometry, electromyography, nerve latency testing, endoanal ultrasound), have failed to provide any benefit in the diagnostic evaluation [34]. Only patients who are candidates for pelvic floor rehabilitative therapy (biofeedback) will benefit from support by anal manometry.

### 2.3. Treatment of LARS

#### 2.3.1. Dietary Modification

In general, dietary modification is regarded as the first-line therapy for patients suffering from LARS symptoms. It has been recommended that patients avoid foods that would lead to soft stools (e.g., caffeine, spicy food, alcohol and fat) [41,42,43]. In addition, intake of high-fiber food should lead to an increase in solid stool consistency, thus improving symptoms or incontinence due to diarrhea.

The positive impact of high dietary fiber was nicely demonstrated in the Nurses’ Health Study of 58,330 women, in which women with the highest intake of fibers (25 g/day) showed a 31% lower risk for fecal incontinence (mainly due to liquid stool) compared with women with the lowest intake [42].

It must be considered seriously that an attempt to increase dietary fiber intake with insoluble fibers could lead to a deterioration of symptoms due to an increased number of bowel movements as well as bloating. Therefore, soluble fiber (bulking agents) should be recommended in order to achieve better stool consistency.

#### 2.3.2. Medication

Medication therapy most commonly focuses on the reduction in colonic motility, and loperamide, as a constipating agent, is the most commonly used treatment [34,43,44]. In addition to the effect of a reduction in defecation episodes, an increase in anal resting tone due to a possible activation of the internal anal sphincter has been reported as the beneficial effect of loperamide.

Recently, 5-HT3 antagonists (e.g., ramosetron) and bile acid sequestrants have shown promising results which will need to be confirmed on a higher level of evidence. Itagaki and coworkers observed an improvement in the Jorge–Wexner incontinence score as well as in the urgency grade and the number of defecation episodes after one month of administrating ramosetron to a cohort of 25 patients after sphincter-saving surgery [45]. However, besides the small number of patients, no randomization has been used.

In accordance with this observation, it should be emphasized that most statements about the efficacy of medication therapy are hampered by the lack of scientific evidence as well as the fact that LARS is associated with a wide variety of different symptoms.

#### 2.3.3. Pelvic Floor Rehabilitation

Few studies evaluating the efficacy of rehabilitative treatment of patients with LARS are available and mostly deal with symptoms of “incontinence” and “stool frequency” [44].

Bartlett and coworkers were able to demonstrate retrospectively a decrease in incontinence and number of defecation episodes in 19 patients who underwent biofeedback and home exercises for four weeks as a treatment for functional problems after colorectal surgery (various procedures) [46].

Although a significant improvement in incontinence scores (compared to baseline values) could be observed, a deterioration was determined after 2 years, with 25% of the patients having forgotten how to perform the training [46].

Similarly, Kim et al. were able to show an improvement in continence function using biofeedback treatment focusing on coordination, sensory function and muscle strength in a retrospectively analyzed cohort of 70 patients [47]. Incontinence function, the number of bowel movements and anal manometry data showed improvement although anal physiology data were only available in 31 of 70 patients.

In a more recent publication, Lee and coworkers evaluated 31 patients suffering from LARS following sphincter-saving surgery to whom biofeedback training (*n*: 16) or supportive therapy (*n*: 15) was offered as one of two therapeutic options [48]. There was no statistically significant difference in LARS scores between both groups. The decrease in the Wexner score and increase in rectal capacity were significantly higher in the biofeedback group. However, it has to be taken into account that this study did not randomize the two groups [48].

A Chinese study by Wu et al. published results of a prospective randomized trial including 109 patients who were allocated to three groups: control, pelvic floor muscle exercise and biofeedback with muscle floor exercise [49]. All patients were followed for 16 months, and data from high-resolution anal manometry, as well as the MSKCC (Memorial Sloan Katering Cancer Center) intestinal function questionnaire, were evaluated. Biofeedback together with muscle exercise led to a significant positive impact on anal physiology data and LARS symptoms compared with mere pelvic floor exercise or control, respectively. Unfortunately, only an English abstract of this publication was available as the full text is published in Chinese [49].

In this context, it is noteworthy that a recently published Australian study protocol (CARRET protocol) dealing with a similar study design has started patient recruitment, which should help provide more insight into the scientific evidence of this treatment approach [50].

#### 2.3.4. Transanal Irrigation (TAI)

TAI has been used previously for functional disorders (incontinence, constipation) of the colon in pediatric patients (spina bifida, anal atresia) as well as in patients following spinal cord injury [50,51]. Based on the successful outcome in these patients, TAI was introduced first in patients suffering from severe and “chronic” (i.e., with a long history) LARS [52].

In a joint Austrian–Swiss study, 14 patients suffering from LARS with a median history of 19 months (9–48 months) and a median number of eight defecation episodes per day (4–12) were treated by TAI [53]. By use of a Foley catheter or the Peristeen irrigation system (Coloplast, Denmark), a median volume of 900 mL (500–1500 mL) of tap water was applied every 24–48 h and led to a significant decrease in defecation episodes as well as improvement in incontinence and QOL scores.

Following this and other similar publications [54], a controlled randomized study was initiated in order to evaluate the efficacy of this approach in an early setting as a prophylaxis against LARS [55]. In this multicenter trial, 39 patients with a protective ileostomy following ultralow rectal resection (median height of anastomosis 3 cm above dentate line) were randomized on the day before planned ileostomy closure either to receive the best supportive therapy against LARS or to start with TAI (1000 mL/24 h) after the first bowel movement. LARS and Wexner scores as well as the SF-36 QOL score were evaluated at 1 week, 1 month and 3 months following ileostomy closure [56].

After one and three months, a significantly better outcome with regards to LARS and Wexner scores could be observed in those patients who regularly performed TAI compared to the group with supportive therapy only, thus showing the positive impact of TAI on the control of LARS symptoms in the early phase after low anterior rectal resection [56].

However, in a follow-up evaluation after 12 months, when patients could freely choose between the two treatment modalities, only 10 of 19 patients who had been allocated to TAI continued with the irrigation therapy. This was noteworthy as TAI patients still revealed significantly fewer defecation episodes and better results in the LARS score compared with the patients in the group with supportive therapy [57].

The main arguments of patients for stopping TAI were the duration of the procedure (approximately 45 min) [57]. Furthermore, it is widely accepted that LARS problems improve over time in many patients, bringing them into a more acceptable situation, as most will not require any special therapy anymore [33].

Although TAI has proven its efficacy in controlling LARS symptoms, a number of questions remain open [33]:a)Volume of irrigation;b)Intervals between irrigation procedures;c)Irrigation devices (Balloon or cone).

There is still an open debate about whether patients should start with a high volume (e.g., 1000 mL) in order to achieve a quick resolution of their symptoms as well as with a regular schedule of TAI every 24 h, or if small volumes (e.g., 100–200 mL) with a significantly shorter toilette time could be equally sufficient to control LARS problems [33].

Furthermore, there is no univocal decision about the optimal irrigation device regarding user friendliness as well as safety considerations.

In general, TAI is a widely accepted and evidence-based option that can be chosen as an effective treatment option for patients with symptoms of LARS. However, it is mandatory to provide patients with a standardized training program supported by experienced and well-trained staff (mostly dedicated stoma therapists) in order to secure appropriate patient compliance (especially in the beginning of the therapy) as well as sufficient safety of the procedure.

#### 2.3.5. Neuromodulation

As already mentioned, it is accepted that surgery and/or radiotherapy can lead to the denervation of the autonomic nervous system and consequently to an impairment of sympathetic as well as parasympathetic nerve fiber activity [20,26]. Damage to these structures can be associated with changes in colonic motility as well as with fecal incontinence.

Therapy aiming to restore neural pathways by continuous neural stimulation has been very successful in treating various forms of urinary and fecal incontinence in the past [55,58].

This can be achieved either by electrical stimulation of sacral nerves at the level of S2 or S3 (so-called sacral nerve stimulation—SNS or sacral nerve modulation—SNM) or the posterior tibial nerve (PTNS) [59,60].

Sacral nerve stimulation (or modulation) is a two-stage procedure, consisting of a test phase and an implant phase [55,58,61].

During the first (test) stage, an electrode is placed at the dorsal root of the sacral nerve S3 or S4 and stimulated externally for two to four weeks in order to evaluate the functional response. Once a positive result (i.e., improvement in symptoms) is confirmed, permanent implant of a neurostimulator is performed subsequently.

While SNS (or SNM) has become the treatment of choice for many patients suffering from fecal incontinence, there is a lack of data dealing with this therapy for LARS patients [55,58]. Although a systematic review showed an improvement in 94% of patients overall (delay of defecation, improvement in QOL scores), it must be accepted that just a few data from 43 patients published in seven studies were available [61].

Due to this relatively low rate of scientific evidence as well as the considerable costs associated with this therapy, SNS cannot be regarded as the first treatment option for LARS.

Contrary to SNS (SNM), tibial nerve stimulation (PTNS) is more easily applicable and significantly cheaper. Percutaneous (needle) or transcutaneous (skin pad) stimulation of the posterior tibial nerve is usually performed in a 30-min session and has been reported in small series with varying results [59,60].

Marinello and coworkers recruited 46 patients for a multicentric randomized trial. Patients with a major LARS score were allocated to receive PTNS or sham therapy (needle placement simulation without nerve stimulation) [62]. The fecal incontinence score was improved after 12 months in the PTNS group (mean(s.d.) score 15.4(5.2) at baseline versus 12.5(6.4) at 12 months; *P* = 0.018). No major changes in QoL or sexual function were observed in either group.

Enriquez-Navascues et al. randomized 27 patients with a major LARS score to receive either TAI or PTNS in a randomized study [62]. The median LARS score decreased from 35 (interquartile range (IQR) 32–39) to 12 (IQR 12–26) (*p* = 0.021) for the TAI group and from 35 (IQR 34–37) to 30 (IQR 25–33) (*p* = 0.045) for the PTNS group.

The Vaizey (incontinence) score fell from 15 (IQR 11–18) to 6 (IQR 4–7) (*p* = 0.037) and from 14 (IQR 13–17) to 9 (IQR 7–10) (*p* = 0.007) with TAI and PTNS, respectively. The authors concluded that both treatments improved the LARS score in this study, but statistical significance could only be reached in the TAI group [63].

## 3. Conclusions

Today, it is widely accepted that following sphincter-preserving rectal resection, a large majority of patients will be affected by LARS symptoms for a period ranging from 12 to 18 months.

Therefore, preoperative information about this potential problem as well as the management of LARS in the postoperative follow-up are equally important as a technically appropriate surgery in order to achieve an acceptable QOL for these patients. Colorectal surgeons, gastroenterologists and nursing staff should be aware of the variety of therapeutic options, and a stepwise approach has been recently suggested by the BOREAL program, as published by Harji and coworkers [42].

Their group proposes an escalating treatment algorithm as illustrated in Figure 1, which was associated with a rate of compliance of 72.9% for the 137 patients who were included. A major LARS decrease from 48% (30 days postoperatively) to 12% at 12 months can be taken as evidence of the effectiveness of the available treatment options to control this functional problem for most of the patients following rectal resection.

## Figures and Tables

**Figure 1 cancers-15-00778-f001:**
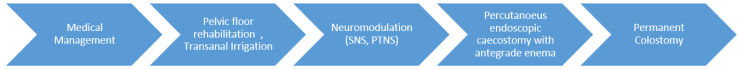
Bowel rehabilitation program (BOREAL) [45].

**Table 1 cancers-15-00778-t001:** Possible risk factors for LARS.

Resection line (low rectum > mid or upper rectum)
Neoadjuvant therapy (radiation > no radiation)
Type of anastomosis (straight coloanal > pouch)
Protective ileostomy > no ileostomy
High ligation > ligation of superior rectal artery

**Table 2 cancers-15-00778-t002:** LARS Symptoms and Consequences [34].

Symptoms	Consequences	Impact on
Unpredictable bowel function	Toilet dependence	Mental and emotional wellbeing
Emptying difficulties	Preoccupation with bowel function	Social and daily activities
Increased stool frequency	Dissatisfaction with bowels	Relationships and intimacy
Repeated painful stools	Strategies and compromises	Roles, commitments and responsibilities
Urgency		
Incontinence		
Soiling		

**Table 3 cancers-15-00778-t003:** Low Anterior Resection Syndrome Score—LARS Score [40].

Do you ever have occasions when you cannot control your flatus (wind)?	
□ No, never □ Yes, less than once per week □ Yes, at least once per week	0 4 7
Do you ever have any accidental leakage of liquid stool?	
□ No, □ Yes, less than once per week □ Yes, at least once per week	0 3 3
How often do you open your bowels?	
□ More than 7 times per day (24 h) □ 4–7 times per day (24 h) □ 1–3 times per day (24 h) □ Less than once per day (24 h)	4 2 0 5
Do you ever have to open your bowels again within one hour of the last bowel opening?	
□ No, never □ Yes, less than once per week □ Yes, at least once per week	0 9 11
Do you ever have such a strong urge to open your bowels that you have to rush to the toilet?	
□ No, never □ Yes, less than once per week □ Yes, at least once per week	0 11 16

No LARS: 0–20; Minor LARS: 21–29; Major LARS: 30–42.

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
