# Peer review of "Management of Low Anterior Resection Syndrome (LARS) Following Resection for Rectal Cancer"

_cancers, 2023, doi:10.3390/cancers15030778_

Round 1
Reviewer 1 Report
Congratulations for this complex review-type article.
Author Response
Thank your for reviewing our paper
Reviewer 2 Report
Comments on abstract:
Please change method to methods. Is it possible to make the results more comprehensive?
Comments on full text:
Introduction: Add a rationale/aim of the study.
Methods: There is no 'methods' section. Please add e.g. which sources are used.
Results:
A table with risk factor of LARS can be illustrative.
Section 4 'treatment of LARS' only contains 4.1 dietary modification. Please change all other treatments (section 5-7) below section 4. Besides, there are 2 section 5's (medication and pelvic floor rehabilitation).
Section 7 (neuromodulation) has a lot of enters in the text & there is a lack of references at the stages explanation part.
A part with future perspective/necessary future research would be from added value.
Conclusion:
A discussion/summary before the conclusion would be from added value. Figure one can be included in this discussion, to make the conclusion shorter and more comprehensive.
Author Response
Thank you for reviewing our manuscript.
We have made all the improvements you suggested and we are grateful for your constructive imput.
Reviewer 3 Report
This narrative review presents a concise and clear report of current literature on LARS. This reviewer does not require changes to the manuscript.
Author Response
Thank you for reviewing our manuscript
Reviewer 4 Report
No notes
Author Response
Thank you for reviewing our manuscript